# Flow-induced elongation of von Willebrand factor precedes tension-dependent activation

Hongxia Fu [1,2], Yan Jiang [1,2], Darren Yang [1,2], Friedrich Scheiflinger[3], Wesley P. Wong [1,2] & Timothy A. Springer [1,2]

Von Willebrand factor, an ultralarge concatemeric blood protein, must bind to platelet GPIbα during bleeding to mediate hemostasis, but not in the normal circulation to avoid thrombosis. Von Willebrand factor is proposed to be mechanically activated by flow, but the mechanism remains unclear. Using microfluidics with single-molecule imaging, we simultaneously monitored reversible Von Willebrand factor extension and binding to GPIbα under flow. We show that Von Willebrand factor is activated through a two-step conformational transition: first, elongation from compact to linear form, and subsequently, a tension-dependent local transition to a state with high affinity for GPIbα. High-affinity sites develop only in upstream regions of VWF where tension exceeds ~21 pN and depend upon electrostatic interactions. Re-compaction of Von Willebrand factor is accelerated by intramolecular interactions and increases GPIbα dissociation rate. This mechanism enables VWF to be locally activated by hydrodynamic force in hemorrhage and rapidly deactivated downstream, providing a paradigm for hierarchical mechano-regulation of receptor–ligand binding.

[1] Program in Cellular and Molecular Medicine, Boston Children's Hospital, Boston, MA 02115, USA. [2] Department of Biological Chemistry and Molecular Pharmacology, Harvard Medical School, Boston, MA 02115, USA. [3] Shire, Vienna 1220, Austria. Hongxia Fu and Yan Jiang, Wesley P. Wong and Timothy A. Springer contributed equally to this work. Correspondence and requests for materials should be addressed to W.P.W. (email: Wesley.Wong@childzrens.harvard.edu) or to T.A.S. (email: Springer_lab@crystal.harvard.edu)

Von Willebrand factor (VWF) is one of nature's longest polymeric proteins, with 2050-residue monomers linked head-to-head and tail-to-tail into concatemers up to 200 monomers in length[1–5]. Each monomer contains multiple domains, including the A1 domain which contains the binding site for platelet GPIbα[5]. VWF is proposed to function as a mechanical flow sensor to activate platelet plug formation by binding to platelet GPIbα in response to bleeding but not in normal circulating blood. Studies of individual VWF domains have revealed that the A1 domain can respond to mechanical forces and interact with GPIbα[6–8]. However, these results do not explain the activation of the full-length VWF concatemer, which is the functional unit in vivo, because A1 domains in VWF concatemers can be shielded by other domains[9–12], and VWF concatemers in blood flow experience force differently from the A1 domains used in these experiments. Bulk VWF in solution or on surfaces can bind platelets above a critical shear stress[3, 13], but

VWF under such conditions may contain a broad distribution of various sizes of concatemers and self-associated bundles[14], which makes it difficult to analyze the molecular mechanism.

The mechanisms by which VWF molecules become activated under flow therefore remain unclear, although two broad hypotheses have been proposed: (1) elongation of the VWF concatemer with unmasking of a constitutively active binding site for GPIbα in the A1 domain that is hidden in the compact conformation or (2) mechanical activation of the A1 domain in the concatemer in response to force[5]. In stasis, single VWF molecules are compact with the concatemeric chain in an irregularly coiled conformation. Theoretically, shear flow can cause polymers to tumble and to alternately extend and contract, and extension increases with increasing shear flow[15, 16]. Although previous studies have suggested that shear flow can induce conformational changes in VWF[17–20], VWF elongation has not yet been observed simultaneously with GPIbα binding.

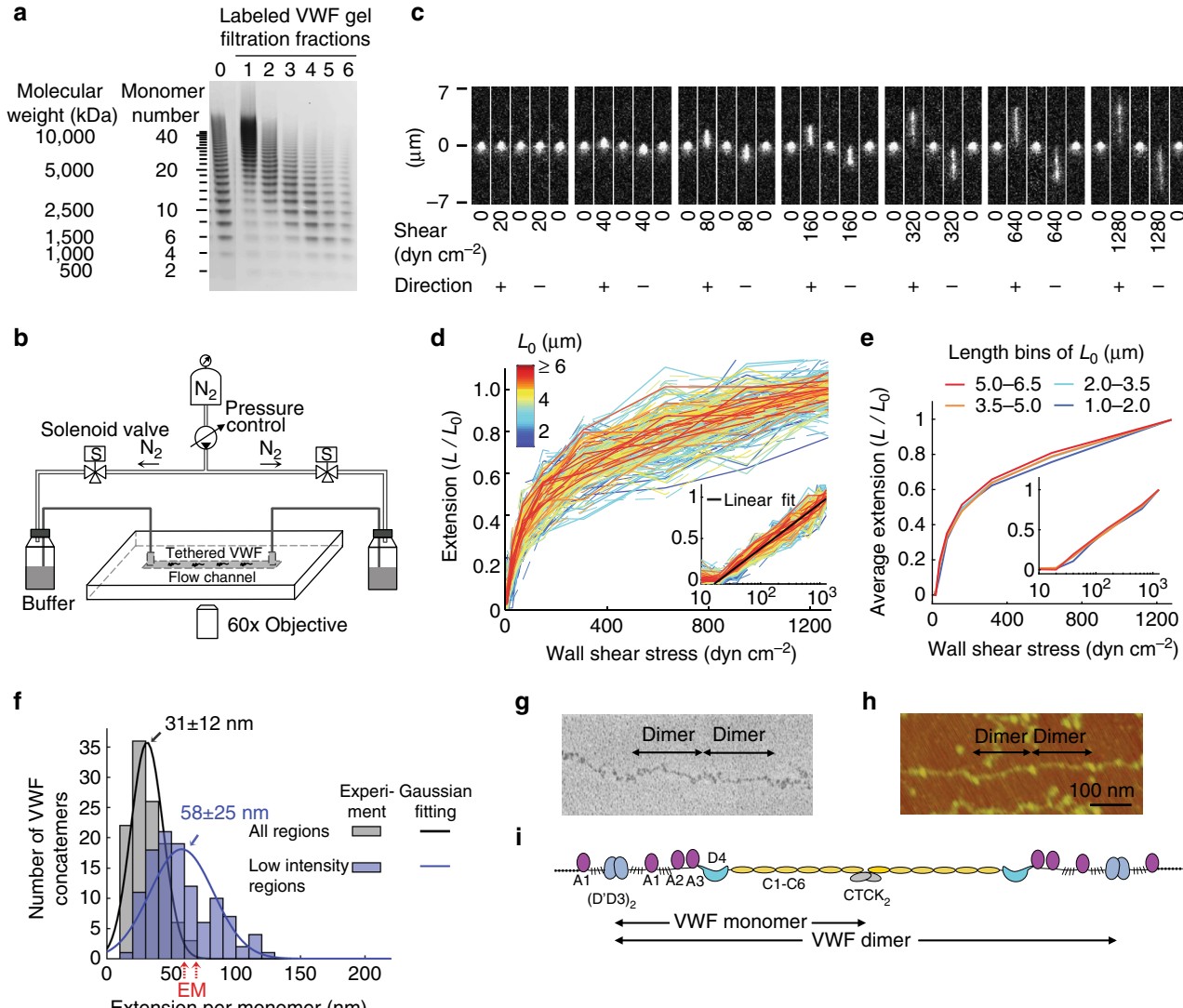

**Fig. 1** VWF elongation under flow. **a** VWF concatemer size analysis by SDS- 1% agarose gel electrophoresis. Lanes are without (0) or with Sepharose CL-2B gel filtration (1–6). **b** Schematic of the TIRF microscope with pressure-actuated flow. **c** Representative VWF molecule under cycles of start- and stop-flow in forward (+) and backward (−) directions at indicated wall shear stresses (10 ms exposure every second). **d** VWF concatemer extension ($N = 112$ concatemers) normalized to length at 1280 dyn cm$^{-2}$ ($L_0$, color key) vs. wall shear stress under forward (*solid lines*) and backward (*dash lines*) flow (Supplementary Fig. 1 a, b). Inset shows the linear fit to log of shear stress. **e** Average VWF concatemer extension in **d** for the four $L_0$ bins. **f** Average extension of VWF monomers for all regions or only the low intensity regions (Supplementary Fig. 2) of VWF concatemers in **d**. *Red arrows*: VWF monomer contour length measured by EM[24, 25]. **g**, **h** EM[24] and AFM[26] images of VWF concatemers. **i** VWF concatemer schematic[5]

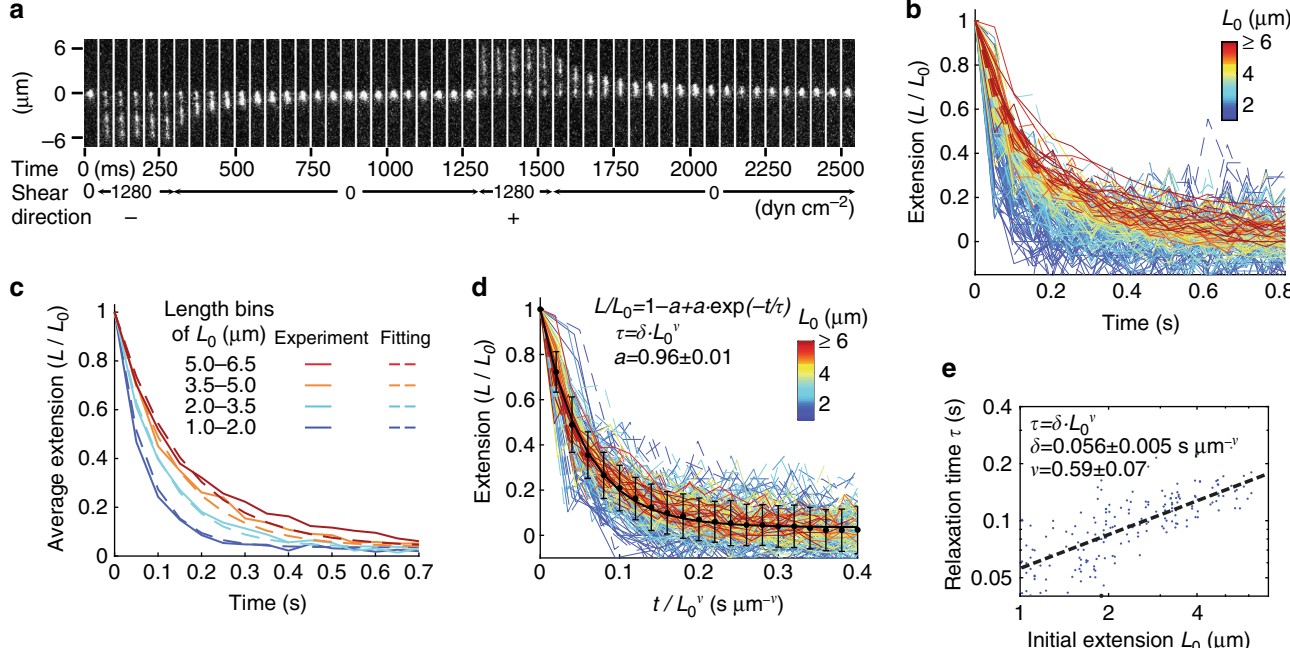

**Fig. 2** VWF relaxation dynamics. **a** Successive 20 frames per second (fps) images of VWF extension in forward (+) and backward (−) flow at 1280 dyn cm$^{-2}$ and relaxation in absence of flow in buffer with 60% w/w sucrose and 150 mM NaCl. **b** Time course of VWF extension normalized to its initial value $L_0$ at 1280 dyn cm$^{-2}$ during relaxation from forward (*solid lines*) or backward (*dashed lines*) flows for 94 VWF concatemers (Supplementary Fig. 1c, d). **c** Average VWF relaxation dynamics (*solid lines*) in **b** for four $L_0$ bins with fits (*dashed lines*) from the equation shown in **d**. **d**, **e** Fits to the equations with 95% confidence intervals shown for VWF relaxation (Supplementary Fig. 1e, f). *Points* and *error bars* show mean and SD for all 94 concatemers relaxed from flow in both directions. The value of 0.96 for *a* suggests that we can account for 96% of the relaxation with a single exponential decay. There may be another component with a slower relaxation, but we could not measure it owing to the photobleaching that would occur during the long exposure required for its measurement

VWF molecules tethered to endothelium or collagen are physiologically important at sites of hemostasis[5]. Tethering abolishes tumbling and lowers the shear stress required for extension of a polymer[15]. To study the activation mechanism of these tethered VWF molecules, we have designed an integrated two-color fluorescence imaging and microfluidic flow system that enables the first simultaneous measurement of flow-induced dynamics of extension/relaxation of single, fluorescent VWF molecules and binding/dissociation of fluorescent GPIbα. Our results reveal an unexpected mechanism that mechanical force activates high-affinity GPIbα binding by driving a two-step conformational transition in VWF. This study thus provides important mechanistic insights into the specific function of VWF in controlling bleeding and, more broadly, into the mechano-regulatory properties of protein-based polymers with complex intramolecular interactions.

## Results

**VWF elongation and relaxation in flow.** Recombinant VWF[21] labeled with biotin (~ 0.3 per monomer) and Alexa Fluor 488 (0.67 fluorophore/monomer) was subjected to Sepharose CL-2B gel filtration. Fraction 1, containing VWF with ~ 20–200 monomers/concatemer (Fig. 1a), was added in stasis to a flow chamber with 1–10 mechanically stable streptavidin[22] molecules per μm$^2$ on the wall. Excess streptavidin sites were blocked with free biotin prior to initiation of flow to prevent multi-site attachment of extended VWF. We were not able to distinguish by fluorescent imaging where along their lengths individual concatemers had bound to the wall; however, binding near ends might have been favored by greater accessibility in the irregularly coiled polymer. VWF on the flow chamber wall was subjected to shear flow

controlled by rapidly switchable gas pressure[23] (Fig. 1b). VWF conformation was monitored in real-time with total internal reflection fluorescence (TIRF) microscopy (Fig. 1b).

In the absence of flow and in low flow, VWF concatemers were compact and globular; at higher flow rates VWF elongated and extension increased with increasing shear stress (Fig. 1c–e, Supplementary Fig. 1a, b). Extension was reversible, as shown in Fig. 1c with many cycles of elongation and compaction of a single VWF molecule as flow was turned on and off, reversed in direction, and increased from 20 to 1280 dyn cm$^{-2}$ (Supplementary Movie 1). Elongation of VWF concatemers with end-to-end distances ranging from 1.0 to 6.5 μm showed similar flow dependence when extension was normalized to the maximal length reached at 1280 dyn cm$^{-2}$ ($L/L_0$, Fig. 1d, e). The observation that fractional extension vs. shear stress does not appear to depend on the length of the concatemers is difficult to explain using standard polymer models and may be dependent on specialized biochemical properties of VWF that are not currently understood. However, we did find that when extension was plotted against the log of shear stress (inset, Fig. 1d), it fit well to a straight line and gave an intercept at 15 dyn cm$^{-2}$. This empirically observed value represents the threshold at which elongation begins, but its physiological significance may be limited, since we find that elongation is not sufficient for VWF activation, as described below.

Extension of single VWF concatemers measured by microscopy at 1280 dyn cm$^{-2}$ was converted to extension per VWF monomer using the fluorophore/monomer labeling ratio and the intensity of single Alexa Fluor 488 fluorophores. Measuring all regions of concatemers including bright, compact regions, or only regions of lower, more uniform fluorescence intensity (Supplementary Fig. 2), gave estimates of 31 ± 12 or 58 ± 25 nm

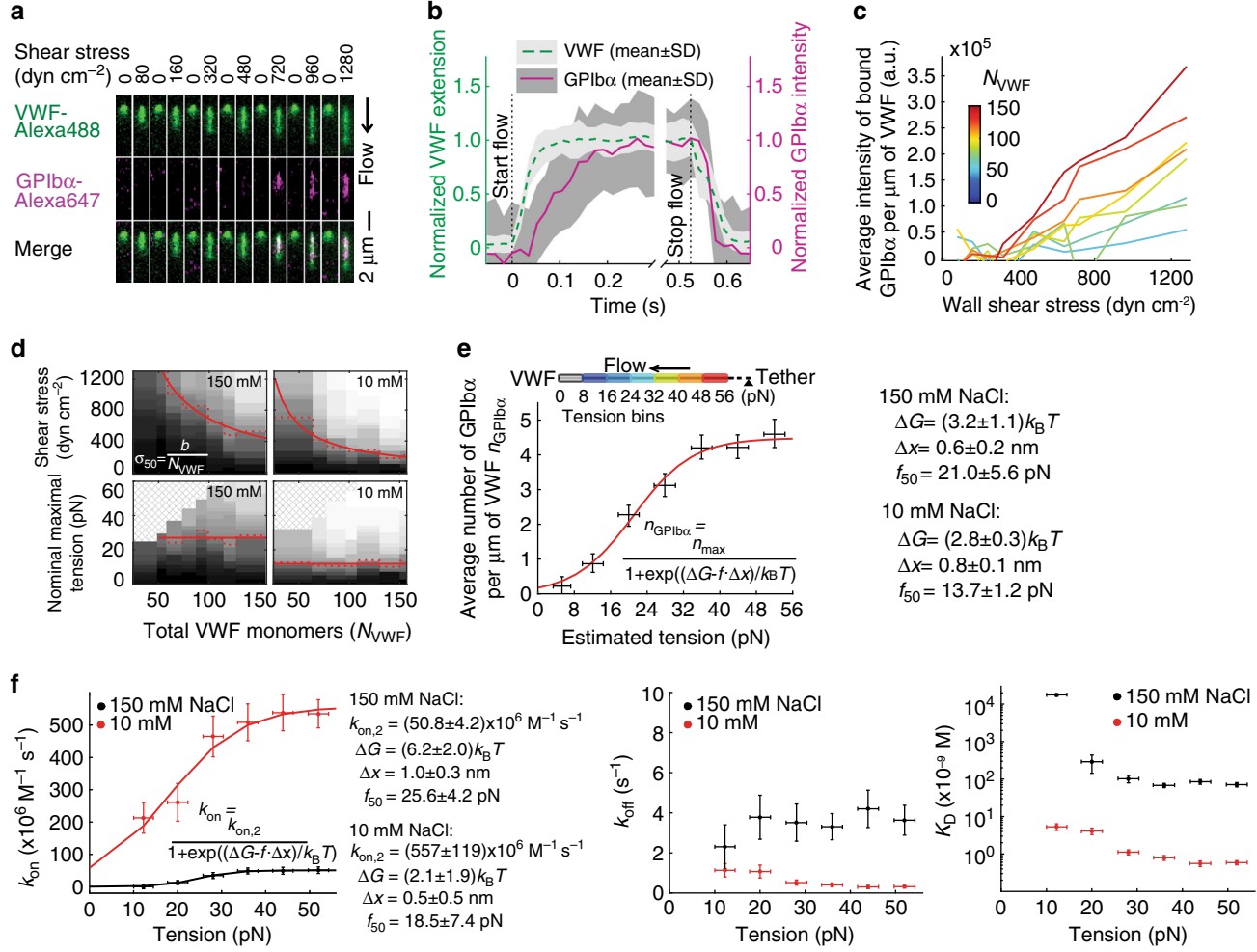

**Fig. 3** Tension-regulated VWF binding to GPIbα. **a** Representative time-lapse dual color fluorescence images of equilibrium VWF extension and reversible binding to GPIbα during cycling between stasis and a range of wall shear stresses. **b** Kinetics (57 fps) of VWF extension/relaxation and binding/ dissociation of GPIbα following flow initiation/cessation at 1280 and 0 dyn cm$^{-2}$, respectively. Data are mean ± SD for 20 concatemers for three flow cycles each, with VWF length and GPIbα binding for each cycle normalized to its average, equilibrium value between 0.37 and 0.52 s after flow initiation. **c** Average fluorescence intensity of bound GPIbα per μm of VWF with increasing wall shear stress. VWF concatemers ($N = 240$) are binned by number of monomers per concatemer ($N_{VWF}$), estimated by total fluorescence intensity. Bins are keyed according to the color bar; see Supplementary Fig. 3 for details. **d** Percentage of activated VWF concatemers by *gray scale* from black (0%) to white (100%) as a function of $N_{VWF}$ (same binning as **c**). Activation is defined as binding over background to GPIbα (Methods). Data in 150 and 10 mM NaCl are with $N = 240$ and 210 concatemers, respectively. Activation is plotted according to wall shear stress (*upper*) or maximal tension (*lower*). *Red lines* show the median (50%) activation in each bin (*dotted*) or fits to the equation shown to all bins (*solid*). Fits to shear stress in 150 and 10 mM NaCl gave *b* values of (6.9 ± 1.2) × 10$^4$ and (3.0 ± 0.2) × 10$^4$ dyn cm$^{-2}$, respectively. **e** Equilibrium binding of GPIbα per μm ($n_{GPIbα}$) in tension bins along the length of VWF concatemers extended at 1280 dyn cm$^{-2}$. Data are mean for 103 concatemers that were long enough to include all seven tension bins. *Red line*: two-state model fit (inset formula). **a–e** Show data with 100 nM GPIbα in 150 mM NaCl; additionally, **d** and **e** (*right*) show data with 20 nM GPIbα in 10 mM NaCl (Supplementary Fig. 3). **a** and **c–e** Average four to six 10 ms exposures taken every 300 ms. **f** Dependence on tension of $k_{on}$, $k_{off}$, and $K_D$ values with 95% confidence interval. Lines show fit of the two-state model to $k_{on}$. Binding was too low in the 0–8 pN force bin to yield meaningful data. Measurements were with 78–141 concatemers at five concentrations of GPIbα in each of 10 and 150 mM NaCl, respectively, at six force bins with kinetics following shift from 0 to 1280 dyn cm$^{-2}$ (Supplementary Fig. 5). $\Delta G$ values in **e** and **f** are within error of one another; uncertainty in **f**, which lacks the data point in the 0–8 pN bin and has higher variance for each data point. In **e**, **f**, error bars are SD for tension and 95% confidence intervals for the other variables

per monomer (mean ± sd), respectively (Fig. 1f). These estimates are within ∼ two fold of the length of linear VWF monomers estimated from electron microscopy (EM), atomic force microscopy (AFM), or the sizes of the component VWF domains[24–27] (Fig. 1g–i). This result suggests that in the extended conformation of VWF, the monomers are linearly arranged along the concatemer, and are no longer free to interact with one another as they were in the loosely coiled conformation. In this linear, extended conformation, hydrodynamic force exerted on a VWF concatemer downstream is transmitted upstream as tensile force along the spine of the concatemer to the tether point. This tensile

force is applied to each domain and to each junction between domains in the force-bearing chain, including the junctions of A1 with mucin-like segments (Fig. 1i).

The dynamics of VWF relaxation following cessation or decrease in flow are important for understanding how its activity can be localized to regions of high hemorrhagic flow in vivo, and also reveal insights into interactions among the domains in the VWF polymer (Fig. 2, Supplementary Fig. 1c–f). To bring dynamics into a measurable range, we increased viscosity 58-fold by adding 60% (w/w) sucrose, which generally has little effect on proteins except to increase the stability of folded domains[28]. Even

**Table 1 VWF-GPIbα binding affinity**

**a**

| NaCl (mM) | | Average tension (pN) | | | | | |
|---|---|---|---|---|---|---|---|
| | | $12 \pm 2.2$ | $20 \pm 2.3$ | $28 \pm 2.3$ | $36 \pm 2.3$ | $44 \pm 2.3$ | $52 \pm 2.3$ |
| 150 | $k_{on}$ ($\times 10^6$ M$^{-1}$ s$^{-1}$) | $0.1 \pm 6.4$ | $13 \pm 6.1$ | $35 \pm 9.3$ | $49 \pm 9.2$ | $49 \pm 11$ | $51 \pm 10$ |
| | $k_{off}$ (s$^{-1}$) | $2.3 \pm 1.1$ | $3.8 \pm 1.1$ | $3.5 \pm 0.9$ | $3.3 \pm 0.7$ | $4.2 \pm 0.9$ | $3.6 \pm 0.7$ |
| | $K_D$ ($\times 10^{-9}$ M) | $(18 \pm 1.3) \times 10^3$ | $290 \pm 150$ | $102 \pm 23$ | $68 \pm 9.0$ | $85 \pm 12$ | $71 \pm 9.0$ |
| 10 | $k_{on}$ ($\times 10^6$ M$^{-1}$ s$^{-1}$) | $213 \pm 47$ | $261 \pm 58$ | $465 \pm 63$ | $508 \pm 57$ | $538 \pm 56$ | $538 \pm 44$ |
| | $k_{off}$ (s$^{-1}$) | $1.1 \pm 0.3$ | $1.1 \pm 0.3$ | $0.5 \pm 0.1$ | $0.4 \pm 0.1$ | $0.3 \pm 0.1$ | $0.3 \pm 0.1$ |
| | $K_D$ ($\times 10^{-9}$ M) | $5.3 \pm 1.1$ | $4.1 \pm 0.8$ | $1.1 \pm 0.2$ | $0.8 \pm 0.1$ | $0.6 \pm 0.1$ | $0.6 \pm 0.1$ |

**b**

| Tension 40–56 pN | 150 mM NaCl | 10 mM NaCl |
|---|---|---|
| $N_{total}$ (/μm) | $7.9 \pm 0.6$ | $6.6 \pm 0.1$ |
| Average distance between binding sites (nm) | $126 \pm 9.9$ | $152 \pm 3.2$ |
| $k_{on}$ ($\times 10^6$ M$^{-1}$ s$^{-1}$) | $50 \pm 11$ | $538 \pm 50$ |
| $k_{off}$ (s$^{-1}$) | $3.9 \pm 0.8$ | $0.3 \pm 0.1$ |
| $K_D$ ($\times 10^{-9}$ M) | $78 \pm 11$ | $0.6 \pm 0.1$ |

(a) Tabulation of data from Fig. 3f. (b) Combined results of fitting all data from Fig. 3f in the two highest force bins ( > 40 pN) where activation by force is maximal. The errors in the table are SD for tension and 95% confidence intervals for the other variables.

in sucrose, VWF extension was rapid and was largely complete in the first 50 ms of movies (Fig. 2a). Relaxation of VWF after flow cessation was slower, at ~300–700 ms (Fig. 2a–c), and followed single-exponential decay dynamics (Fig. 2d, Supplementary Movie 2). Relaxation took longer for longer concatemers, as evident when VWF molecules were binned by length (Fig. 2c). Fitting the relaxation of all individual concatemers to exponential decay gave a characteristic relaxation time, $\tau$, that scales with initial length $L_0$ to the power $\nu$ (Fig. 2e). Normalizing time with $L_0^\nu$ allowed all relaxation curves to be described by a single formula (Fig. 2d). The scaling laws represented by the equations shown in Fig. 2d hold for a wide range of synthetic chemical polymers[29, 30]. The finding that VWF follows the same laws allows prediction of the behavior of a VWF concatemer regardless of its size, and also enables comparison of VWF to other polymers. For well-solvated polymers that relax solely due to diffusion of their flexible segments, $\nu$ can be calculated to be 1.5–1.8 using the Zimm model, in agreement with experimental measurements[29, 30]. In contrast, the exponent $\nu$ we measured for VWF is far lower: $0.59 \pm 0.07$ (Fig. 2e), which makes the kinetics of relaxation much less length dependent. Furthermore, the low value of $\nu$ strongly suggests that attractive interactions among the modules within individual VWF single molecules greatly accelerate the relaxation of longer concatemers to the compact conformation. Thus, VWF is biologically specialized for rapid compaction once it is no longer exposed to high flow.

**Tension regulates affinity of VWF for GPIbα.** To study activation of VWF by flow, we added Alexa Fluor 647-labeled GPIbα to the system and measured VWF extension and GPIbα binding by two-color fluorescence (Fig. 3). In cycles of stasis and flow with increase in flow in each cycle, GPIbα binding was only observed at 720 dyn cm$^{-2}$ and above for the VWF molecule of the length shown in Fig. 3a. Cessation of flow immediately reversed GPIbα binding (Fig. 3a, b, Supplementary Movie 3). Notably, extension was not sufficient for GPIbα binding: extension but not GPIbα binding occurred at 80–480 dyn cm$^{-2}$ (Fig. 3a). Moreover, at 720–1280 dyn cm$^{-2}$, binding occurred to upstream regions of the VWF molecule near the tether where tensile force is high, but not to downstream, extended regions where force is lower (Fig. 3a).

Lack of GPIbα binding to downstream ends of concatemers and dependence of binding on VWF size (Fig. 3c) suggest that it is not shear stress per se, which is uniform along the length of the molecule, but the cumulative effect of this shear stress along the length of a VWF concatemer in building up tension along its spine that activates binding. To further test this concept, we measured whether shear stress or tensile force sets the threshold for activation. We first used a method that was applicable to a wide range of shear stresses and concatemer lengths, and which used the entire VWF concatemer as the length, since it did not resolve the position of the tether point along this length (Fig. 3c, d). GPIbα bound to VWF only above a size-dependent threshold shear stress $\sigma_T$ (Supplementary Fig. 3a). The shear stress at which 50% of VWF concatemers were activated ($\sigma_{50}$) was inversely correlated to the total number of monomers in each VWF concatemer ($N_{VWF}$) so that $\sigma_{50} = b/N_{VWF}$ (Fig. 3d, *upper*). On the other hand, when activation was plotted as a function of tension, a uniform threshold was found independent of $N_{VWF}$ (Fig. 3d, *lower*). We also determined $\sigma_{50}$ values at low ionic strength (10 mM NaCl), which increased VWF affinity for GPIbα (Fig. 3f, Table 1) but did not alter the shear dependence of VWF extension or the kinetics of relaxation (Supplementary Fig. 4a, b). Strong ionic dependence was consistent with the high electrostatic complementarity of the A1–GPIbα interface[31] and is relevant for facilitating rapid binding of GPIbα to VWF (Discussion). The threshold for activation of VWF was ~50% lower in 10 mM than in 150 mM NaCl by both measures (Fig. 3d). Importantly, the direct relationship between the nominal tension required for activation, both over a range of shear stresses and at two ionic strengths (Fig. 3d, *lower*) provided strong support for the concept that tensile force, rather than shear stress, is the essential factor that regulates the activation of VWF concatemers for GPIbα binding.

We next quantitated the relationship between force and GPIbα binding along the lengths of single, long VWF concatemers between the tether point and free end (Fig. 3e). Tensile force within VWF concatemers is proportional to the number of monomers downstream and thus is maximal at the upstream, tethered end and zero at the downstream, free end (Fig. 3e, *upper*) and was estimated to be 0.5 pN per monomer at 1280 dyn cm$^{-2}$ (Methods). The number of bound GPIbα molecules per μm of VWF showed sigmoidal dependence on tensile force (Fig. 3e).

Binding was modest below 10 pN, increased greatly between 10 and 35 pN, and plateaued above 35 pN (Fig. 3e). This behavior matches a simple two-state model: tensile force promotes a switch from low-affinity state 1 of the A1 domain in VWF to high-affinity state 2. Indeed, the data fit well to a model in which force alters the free energy difference $\Delta G$ between the two states by $-f \cdot \Delta x$ (*red curve* in Fig. 3e) and in 150 mM NaCl yielded a difference in energy between the two states in the absence of force of $3.2 \pm 1.1$ $k_B T$, a length parameter $\Delta x$ of $0.6 \pm 0.2$ nm, and half-maximal activation at a force of $21.0 \pm 5.6$ pN. Fit to the binding data in 10 mM NaCl yielded a similar $\Delta x$ of $0.8 \pm 0.1$ nm, a $\Delta G$ of $2.8 \pm 0.3$ $k_B T$, and half-maximal activation at $13.7 \pm 1.2$ pN (Supplementary Fig. 3d).

**Rapid binding kinetics.** Speed is essential to hemostasis in the high pressure arterial circulation and we therefore extended our measurements to kinetics. Using multiple concentrations of GPIbα and fitting force-binned GPIbα intensity kinetics traces upon flow initiation to Eq. (6) in Methods, we determined force-dependent $k_{on}$ and $k_{off}$ values (Fig. 3f, Table 1, Supplementary Fig. 5). To test for an electrostatic contribution to on-rate[32], we compared kinetics in 10 and 150 mM NaCl. For a protein–protein interaction, $k_{on}$ was unusually fast in 150 mM NaCl at $50 \times 10^6$ $M^{-1}$ $s^{-1}$. Furthermore, $k_{on}$ was ~10-fold faster in 10 mM NaCl, demonstrating strong electrostatic steering (Table 1). $k_{off}$ was also ~10 times lower in 10 mM NaCl, likely from electrostatic attraction as well. Consistent with the two-state

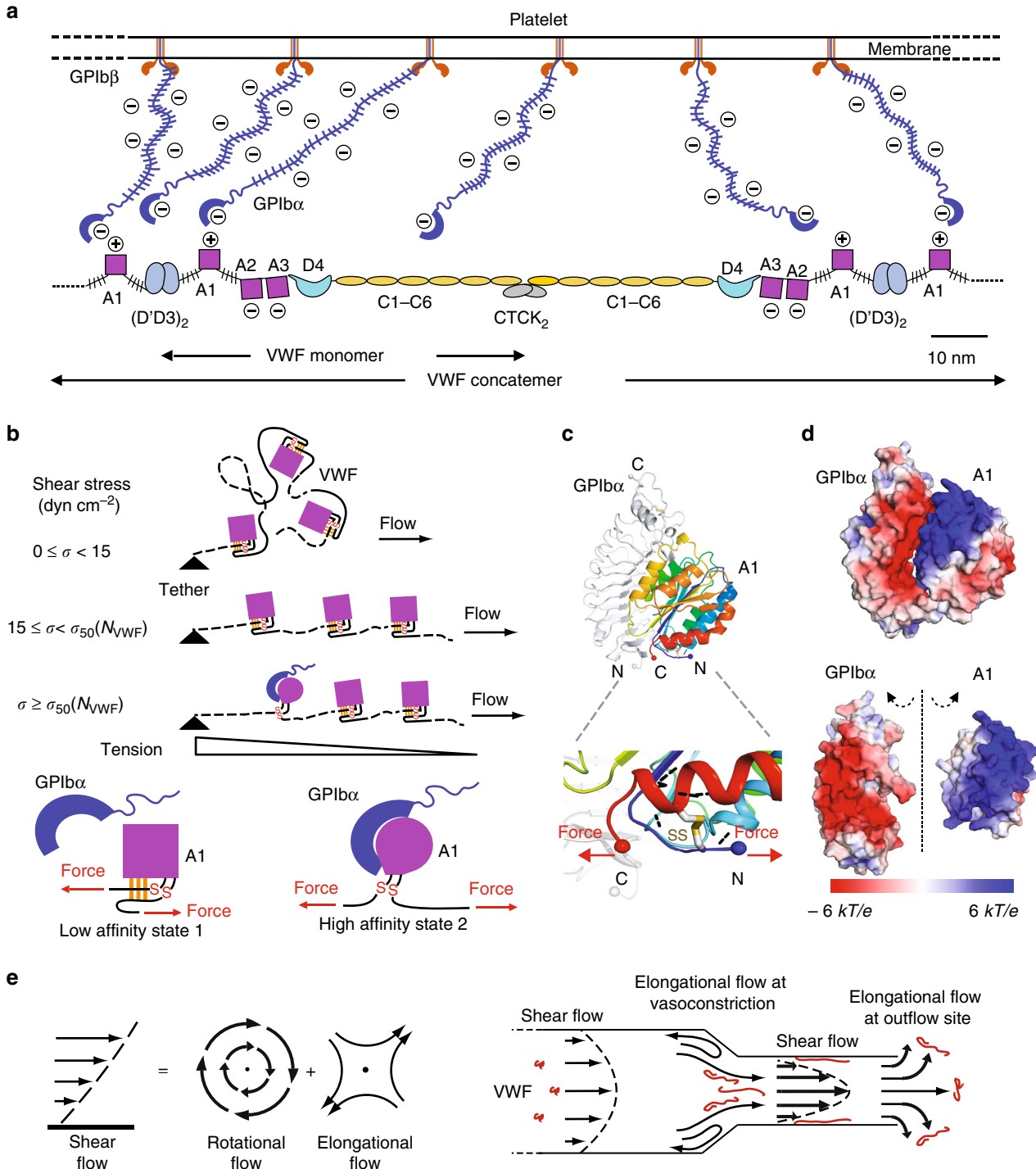

model, force increased $k_{on}$ similarly to the increase in population of state 2. The increase in $k_{on}$ with force and its plateau also fit to the two-state model (lines in Fig. 3f), although with larger errors. In the presence of saturating tensile force, i.e. above 40 pN, state 2 of VWF has a $K_D$ for GPIbα of ~80 nM as estimated from $k_{off}/k_{on}$. The number of GPIbα molecules bound per μm of VWF at equilibrium was ~8, or 1 molecule per ~126 nm (Table 1). At the high level of tensile force at which GPIbα binding was measured, A2 domain unfolding may increase VWF length per monomer (Discussion). Furthermore, the fluorescence intensity of GPIbα bound to VWF will be lower than that of GPIbα bound to the substrate used in calibration, owing to the decrease in intensity of the TIRF field (Methods).

## Discussion

Our measurements of VWF elongation and activation provide key insights into its hemostatic function (Fig. 4). Furthermore, these measurements connect VWF length scales measurable by light microscopy to atomic resolution studies of domain structure and EM and AFM measurements of domain arrangement in VWF monomers and dimers (Fig. 1g–i). Mature VWF monomers contain a large D'D3 assembly at their N-terminus that forms the head-to-head dimer interface, three moderately sized VWA domains (A1, A2, and A3) with a mucin-like flexible segment flanking A1 on each side, a D4 assembly, six small, elongated VWC domains, and a small C-terminal domain that forms the tail-to-tail dimer interface (Fig. 4a). We estimated the length per monomer of VWF from its fluorescence intensity when elongated at 1280 dyn cm$^{-2}$. The molecular substructure of VWF and a length per monomer of 60 nm had previously been deduced from EM images of VWF[24] (Fig. 1g) that had been extended and combed by the surface tension at a receding meniscus[33]. A similar length per monomer was determined by AFM imaging of VWF adsorbed to substrates at 30,000 s$^{-1}$[26] (Fig. 1h). The sizes of the domains of VWF and the lengths of the mucin-like segments that flank the A1 domain measured by EM were added together in another study to yield an estimate of 70 nm per monomer[25]. Our estimates of $31 \pm 12$ or $58 \pm 25$ nm per monomer from fluorescence microscopy are sufficiently similar to these prior estimates of monomer length to allow us to conclude that VWF is largely converted from an irregularly coiled to a linear conformation when tethered to a vessel wall at 1280 dyn cm$^{-2}$; however, several uncertainties should be noted. A2 domain unfolding would increase the length per monomer by up to ~50 nm[34, 35]. Alternatively, a recently noted interaction between D4 domains within VWF dimers at neutral pH would shorten monomer length by ~40 nm[27]. Nonetheless, we may conclude from our estimate of

length per monomer that VWF is uncoiled; furthermore, it is important that in all of the above scenarios, the A1 domain would be in the force-bearing concatemer spine, suspended between its flanking mucin-like segments, and elongation would break most if not all of its interactions with other domains (Fig. 4a). The similar average distances between A1 domains in elongated VWF and between GPIb molecules on the platelet surface (Fig. 4a) facilitate multi-point attachment of VWF to platelets. Although this first transition of elongation is not sufficient to activate adhesiveness, it is necessary, since weak interactions between distal parts of the VWF concatemer must be broken to re-route tension and enable it to build up along the central concatemer spine. When tension reaches ~21 pN, it mechanically induces a second transition of A1 to a high-affinity conformation. This transition is localized within the concatemer spine to upstream regions that experience tensile forces of >21 pN and is the key step for activating A1 for binding to GPIbα (Fig. 4b). Length is essential for hemostasis by VWF and shortened concatemers can cause both inherited and acquired forms of von Willebrand disease[5]. Our results show that length is not only important to make VWF multimeric, but also is essential for mechanical tension to reach the level required for activation.

Our data suggest that A1 has two states: a flexed, low-affinity state 1 and an extended, high-affinity state 2 with a longer distance between the N and C termini of A1 to which mechanical tension is applied (Fig. 4b). Fits to this two-state model suggest a difference in length between the states on the order of 0.6–0.8 nm. Intriguingly, this measurement is consistent with a model of A1 affinity regulation based on A1 and A1–GPIbα crystal structures, and the localization of von Willebrand disease gain of function mutations in the A1 domain near its long-range disulfide bond[5–7]. Specifically, A1 contains residues that are external to its long-range disulfide bond that form hydrogen bonds to residues internal to the disulfide bond (that lie between the two cysteines in sequence) and that are hypothesized to stabilize state 1 (Fig. 4c, lower). Force-induced transition of the polypeptide backbone of these residues to an extended polypeptide backbone conformation breaks the hydrogen bonds to the internal residues and results in an extension of up to 1.4 nm[7], consistent with the fit to the two-state model here.

The plateau in the affinity of state 2 above ~35 pN is consistent with the disulfide bond preventing any further effect of force on A1 conformation, and fits expectations for a flex-bond[6, 7]. Flex-bonds have two states: a low-affinity flexed state and a high-affinity extended state. Flex-bond behavior was previously revealed with a single-molecule construct in which A1 was fused through a polypeptide chain linker to GPIbα and force was applied both across the receptor–ligand bond and across A1.

**Fig. 4** Molecular model of flow-induced VWF activation. **a** Schematic of extended VWF[5] and platelet surface drawn to scale, including average spacing between GPIbα/GPIbβ 1:2 complexes and length of mucin-like regions shown as brush-like curves. "+" and "−" indicate net electrostatic charge at plasma pH; sialic acid makes mucins negatively charged. **b** Model of VWF activation under flow. VWF is tethered on a vessel wall. Three values of shear stress (σ) are shown that do not extend, extend but do not activate, or extend VWF and are above the threshold shear stress required for activation ($\sigma_{50}$, a function of $N_{VWF}$). Tension is proportional to the number of VWF monomers downstream of the tether point. Activation of A1 is schematized as disruption by mechanical tension of hydrogen bonds involving residues external to the A1 disulfide, converting A1 from a low-affinity square shape to a high-affinity round shape. **c, d** Views of the GPIbα–A1 complex crystal structure[55] with similar orientations in the upper portion of each panel. **c** Ribbon cartoons of GPIbα in *gray* and A1 in rainbow from N (*blue*) to C-terminus (*red*). Spheres mark N and C-termini. The lower panel shows the force-bearing region at the N and C-termini near the long-range disulfide (*gold stick*). Hydrogen bonds involving residues external to the disulfide are shown as *black dashes*. **d** Electrostatic surface potentials colored according to the key. In the lower, open-book view, GPIbα and A1 are rotated 90° towards the viewer around the dashed axis to show their highly electrostatic interfaces. **e** Shear and elongational flows[5]. (*Left*) Shear flow can be represented as the combination of elongational flow and rotational flow. Arrows show flow streamlines and dots indicate no-flow regions. (*Right*) Effects of vasoconstriction and bleeding on flow. VWF concatemers (*red*) are more compact in shear flow, and as the elongational component of flow increases, they are more extended. When tethered to the vessel wall, as shown in the constriction site, VWF is more easily extended when tethered than when in free solution since tension exerted on it cannot be minimized by movement with the flow—rather, the tension in the molecule must resist the total drag force induced by the flow on the downstream portions of VWF

Switching to the high-affinity state occurred at forces >10 pN and was hypothesized to occur as a consequence of force exertion across the A1 domain rather than across the A1–GPIbα receptor–ligand bond[7]. Our results suggest that tensile force exerted across the A1 domain within a VWF concatemer is indeed sufficient to induce high affinity of A1 for GPIbα. A contrasting catch-bond model places the force not across A1, but across the receptor–ligand bond between A1 and GPIbα, and proposes that increasing tensile force first increases and then decreases affinity[8]. We did not apply force between A1 and GPIbα and observe no such decrease. The stability of the high-affinity state to increasing force makes tensile force exerted along the length of VWF concatemers a robust mechanism for activating hemostasis over a wide range of flow environments and vascular beds in vivo.

Interestingly, the critical tension required to activate the A1 domain is in a similar range to the tension required for unfolding the A2 domain[34, 35]. Unfolding of A2 exposes it to cleavage by the constitutively active plasma protease ADAMTS13. Thus, under physiological conditions, binding of platelets through GPIbα to tension-bearing VWF concatemers and cleavage by ADAMTS13 are competing processes that regulate hemostasis and thrombosis[5]. Furthermore, hydrodynamic drag on platelets will be transmitted to the VWF to which they are bound, increase the tensile force on the VWF, and further enhance both platelet binding and cleavage by ADAMTS13[36, 37]. Moreover, limiting the length of VWF by ADAMTS13 cleavage provides a mechanism for limiting the tension that can be applied to tethered VWF.

Our measurement of GPIbα affinity and kinetics with VWF elongated in flow places these measurements in a much more physiologic context than previously possible[7, 11, 38, 39]. The affinity and on-rate are higher than previously measured, no doubt in part because previous measurements are on ensembles containing both states 1 and 2 of A1. Based on the free energy difference between the states in 150 mM NaCl and the Boltzmann distribution, 5% of A1 will be in the high-affinity state and the apparent affinity measured for the ensemble in stasis will be ~5% of that measured here for state 2. Notably, in the compact conformation of VWF present in stasis and at lower shear stresses, those 5% of A1 domains will largely be inaccessible for binding to platelets. Thus, both elongation of VWF with exposure of all of its A1 domains for binding to platelets and mechanical activation of A1 domains by tensile force within concatemers contribute to making VWF an extremely sensitive switch for hemostasis.

The on-rate of GPIbα to tethered VWF of $10^7$–$10^8$ $M^{-1}$ $s^{-1}$ is faster than typically found for protein–protein interactions ($10^5$–$10^6$ $M^{-1}$ $s^{-1}$) and in a range associated with biological processes where speed is essential[32]. The increase in on-rate with decrease in ionic strength demonstrates that the electrostatically complementary A1 and GPIbα binding interfaces[31] (Fig. 4d) are functionally important to guide the encounter trajectory of GPIbα with A1[40, 41]. Electrostatics must also be important in the context of encounter between elongated VWF and the surface of a platelet (Fig. 4a). GPIbα is tethered to the other subunits of the GPIb complex through a hydrophilic stalk, which contains a long mucin-like segment and shorter segment with sulfated tyrosines proximal to the leucine-rich repeat A1-binding domain. The platelet surface and stalk each are negatively charged and thus repulsion will extend the negatively charged A1-binding domain far from the platelet surface for exposure to VWF. Among the domains in VWF, only A1 is strongly positively charged at physiologic pH, and therefore will orient toward the platelet more than other VWF domains (Fig. 4a). This distribution of electrostatic charge and the flexibility of mucin-like segments are likely to enhance VWF-platelet binding kinetics in vivo.

All of our measurements are on VWF tethered to the wall of a vessel, which occurs in vivo when VWF is secreted from endothelial cells or when VWF free in the circulation binds to collagen at a site of vessel injury[5]. VWF is more easily extended when tethered than when in free solution since tension exerted on it cannot be minimized by movement with the flow—rather the tension in the molecule must resist the total drag force induced by the flow on the downstream portions of VWF (Fig. 4e). In contrast, VWF free in shear flow is subject to rotation and alternating cycles of elongation and compression, with the tension experienced dependent not on the total fluid velocity, but only on the difference in velocity between different sides of the molecule (Fig. 4e). Therefore, much higher wall shear stresses would be required to elongate and activate VWF free in the circulation than for the tethered VWF studied here[5]. Furthermore, our findings for tethered VWF suggest that the shear stress required for VWF extension in free shear flow may be higher than previously suggested[19].

In hemorrhage, blood flow increases up to sixfold in proximal arterioles[42] and together with vasoconstriction[43] (Fig. 4e) increases wall shear stress up to 12-fold. The increase from up to about 50 dyn cm$^{-2}$ in normal arterioles[44, 45] to as high as ~600 dyn cm$^{-2}$ in injured arterioles brings the tensile force on a tethered concatemer with >100 monomers into the range sufficient for A1 domain activation (Fig. 3d, upper). In arterial stenosis as in atherosclerotic injury, a 90% reduction in arterial cross section would similarly increase shear stress on tethered VWF and could cause thrombosis[46]. At sites of vasoconstriction and stenosis, shear flow is altered to elongational flow, and this will also increase the tendency of VWF free in flow to elongate and experience tensile force (Fig. 4e). Furthermore, the binding of platelets to VWF concatemers will increase the tensile force applied in VWF[36, 37], which may lower the critical shear stress for VWF activation.

We found that upon flow cessation, VWF concatemers rapidly relaxed to their compact, irregularly coiled conformation. Relaxation was also associated with rapid dissociation of bound GPIbα. The relaxation dynamics of VWF concatemers was length dependent with power-law scaling, but with an exponent $\nu$ for the characteristic relaxation time $\tau$ that was much lower than for well-solvated chemical polymers[29, 30]. This finding suggests that long-range interactions occur within a VWF concatemer that enable rapid compaction once it is no longer exposed to high flow. These interactions do not appear to be electrostatic, because relaxation kinetics were essentially identical in 10 and 150 mM NaCl. The nature of the long-range interactions within VWF concatemers that specialize them for rapid relaxation remain to be defined; they may be similar in nature to interactions that enable association between VWF concatemers in high flow, which also remain to be defined in terms of the specific domains required[14].

It is important that VWF activation be highly localized, because many vessels may both bleed and send some of their flow to the downstream circulation. Tethered VWF will be cleaved by ADAMTS13 and large fragments will flow downstream[35, 47]. While the off-rate for high-affinity state 2 of A1 is ~3 s$^{-1}$, upon flow cessation, GPIbα dissociates with an off rate $\geq 23 \pm 16$ s$^{-1}$ (Fig. 3b), suggesting rapid transition to and then dissociation from state 1. Fragments released into shear flow will tumble and rapidly relax, and their relaxation may be approximated by that of tethered VWF subjected to flow cessation. Given these kinetics and a blood transport rate of ~7 mm s$^{-1}$ in arterioles[42], we estimate that dissociation of 90% of GPIbα occurs within 1 mm downstream from the injury site. Thus, tension-regulated conformational change in VWF ensures that platelet adhesion can only occur in highly specific locations in the vasculature

where damage to the vessel wall or high flow suggests that bleeding is under way, while adhesive function is rapidly turned off downstream to minimize the likelihood of thrombus formation in patent vessels.

## Methods

**VWF and GPIbα.** Recombinant human VWF expressed in Chinese hamster ovary cells in serum and protein-free media was captured on an ion exchange column, treated with furin to remove incompletely processed pro-domains, solvent and detergent treated, and purified to >99% by two further chromatography steps in a proprietary process (Baxter BioScience, Vienna, Austria)[21]. VWF (0.95 mg ml⁻¹, 315 µl) was labeled with sulfosuccinimidyl-6-[biotin-amido] hexanoate (EZ-link Sulfo-NHS-LC-Biotin, Thermo Fisher Scientific, Waltham, MA, USA) (14 µl at 2.2 mM) in 150 mM NaCl and 20 mM HEPES (pH 7.4) for 1 h at 22 °C and subsequently with Alexa Fluor 488 sulfodichlorophenol ester (Thermo Fisher Scientific) (9.2 µl at 5.82 mM) in the above solution plus 49 µl 1 M NaHCO₃ for 1 h at 22 °C. Labeled VWF was size-fractionated and separated from free biotin and Alexa Fluor 488 on a Sepharose CL-2B (Pharmacia Fine Chemicals, Sweden) column (1 cm × 30 cm) in 150 mM NaCl, 20 mM HEPES pH 7.4, 0.02% Tween 20. Alexa Fluor 488 concentration in each fraction was calculated from fluorescence intensity using Fluorolog-3 (Horiba Scientific, Edison, NJ, USA) and Alexa Fluor 488 stock solution in same buffer as a reference. VWF concentration was calculated from absorbance at 280 nm after correction for Alexa Fluor 488 absorbance. Gel electrophoresis of VWF multimers was as previously described[48, 49]. The gel was fixed on GelBond film (GE Healthcare, Chicago, IL USA), incubated with polyclonal rabbit anti-human VWF (A0082, Dako, Agilent Technologies, Carpinteria, CA, USA) (1:3000), washed in Tris-buffered saline with 0.05% Tween 20 (TBST), and incubated with donkey anti-rabbit IgG (H+L) Alexa Fluor 488 (A-21206, Thermo Fisher Scientific) (1:1000). VWF multimeric bands were detected by FUJIFILM FLA-9000 image scanner (Fujifilm Life Science, Stamford, CT, USA).

The cDNA of wild-type human platelet GPIbα (His¹ to Arg²⁹⁰) was cloned into the ET8 vector[25] with a C-terminal His₆ tag and transiently transfected into HEK293T cells (ATCC, Manassas, VA, USA) using polyethylenimine (Sigma-Aldrich, St. Louis, MO, USA). Cell line was verified free of mycoplasma contamination using mycoplasma detection kit (Lonza Biologics, Portsmouth, NH, USA). Culture supernatants in FreeStyle 293 medium (Thermo Fisher Scientific) were harvested 3 days after transfection and proteins were purified using Ni-NTA affinity chromatography (Qiagen, Valencia, CA, USA) followed by size-exclusion chromatography (Superdex 200 10/300 GL, GE Healthcare Life Sciences) in 150 mM NaCl and 20 mM HEPES (pH 7.4)[6]. Alexa Fluor 647 NHS Ester (Thermo Fisher Scientific) (103 µM) was conjugated to GPIbα (1.5 mg ml⁻¹) in the above buffer with NaHCO₃ added to a final concentration of 130 mM for 1 h at 22 °C. Free dye was removed by dialysis in 150 mM NaCl, 20 mM HEPES (pH 7.4) at 4 °C. Concentrations of GPIbα and Alexa Fluor 647 were calculated from A280 and A650 measurements (NanoDrop, Thermo Fisher Scientific). The molar ratio of fluorophore: GPIbα used in experiments was 1.01:1. VWF and GPIbα samples were stored in aliquots at −80 °C.

Flow experiments were in 150 mM NaCl, 20 mM HEPES (pH 7.4), 0.02% Tween 20, 0.1 mM D-biotin (Invitrogen, Carlsbad, CA, USA), 0.5 mg ml⁻¹ BSA together with 2.2 mM protocatechuic acid (Santa Cruz Biotechnology, Santa Cruz, CA, USA) and 37 nM protocatechuate-3,4-dioxygenase (Sigma-Aldrich) as oxygen scavengers[50], and contained only when noted 60% (w/w) sucrose or 10 mM rather than 150 mM NaCl. For fine tuning the driving pressure of the flow system in experiments that included measurements below 80 dyn cm⁻², 2.5 × 10⁻⁶ w/v 1 µm diameter polystyrene particles (PP-10-10, Spherotech, Lake Forest, IL, USA) were added as flow tracers. In total, 60% (w/w) sucrose was used to increase viscosity 58-fold to 58 cP in VWF relaxation experiments.

**Shear-stress control and dual-color TIRF imaging system.** To construct the microfluidic flow channel, low-density biotin-PEG/PEG-coated #1.5 cover glass (Bio 01, MicroSurfaces, Englewood, NJ, USA) was overlaid with a central piece of ~100 µm thick, double-sided Kapton polyimide tape with 0.5 mm × 15 mm cutouts and topped with a 1.1 mm thick glass slide with two access holes for each channel. A ~3 mm thick polydimethylsiloxane slab with 0.75 mm diameter holes was aligned with the access holes and clamped onto the glass sandwich. A mixture (1:1) of BSA (BSA-Block, Candor, Germany) and casein (The Blocking Solution, Candor, Germany) was added for 30 min, then Traptavidin (0.2 µg ml⁻¹) (Kerafast, Boston, MA, USA) was added for 10 min, and after washing, biotin- and Alexa Fluor 488-labeled VWF concatemers (1 µg ml⁻¹) were added for 10 min (Fig. 1a). A concentration of 5 mM D-biotin was then added for 20 min to block free Traptavidin. The holes on each end of the flow channel were connected through 1/32″ outer diameter PEEK tubing (1569, IDEX Health & Science, Middleborough, MA, USA) to ~1 ml of buffer in a 4 ml gas tight vial containing nitrogen gas (Fig. 1b). Each vial was further connected to a three-way solenoid valve (411L3312HV, ASCO Valve, Fort Mill, SC, USA) to rapidly switch the gas between atmospheric pressure and high pressure gas supplied by a high-precision electronic pressure regulator (DQPV1TFEE010CXL, Proportion-Air, McCordsville, IN, USA) connected to compressed nitrogen cylinder (>99% pure, Lifegas, Marlborough,

MA, USA) to control the flow rate (Fig. 1b). Flow channels could be re-generated by several cycles of rapid injections of 50 µl of 5 mM D-biotin in 150 mM NaCl, 20 mM HEPES (pH 7.4) followed by air through the channel using a 1 ml syringe until no fluorescent VWF was detected by TIRF microscopy.

Wall shear rate was calculated from the volumetric flow rate and the height and width of the flow channel[51]. The volumetric flow rate was measured by weighing the buffer bottle before and after 15–30 s of buffer flow driven at 1.2, 3, and 6 psi. As expected, the volumetric flow rate was approximately proportional to pressure. The measured slope of the linear relation was used to calculate the flow driving pressure to attain any given shear rate. For buffer with 60% (w/w) sucrose, the volumetric flow rate was measured with 1 min of buffer flow driven at 2 and 4 psi. The volumetric flow rate for 60% sucrose buffer was ~1/58 of aqueous buffer at the same driving pressured as predicted by its 58-fold higher viscosity, i.e., the same wall shear stress was induced by the same driving pressure regardless of buffer viscosity. Approximately, the highest shear stress we could apply to the micro channel without deforming the #1.5 cover glass was 1280 dyn cm⁻² . Above this level, the fluorescent imaging suffered from severe defocusing when flow started or stopped.

Fluorescence microscopy used a dual-color TIRF microscope built on a vibration damping optical table (RS4000, Newport, Irvine, CA, USA) with a 60× oil TIRF objective (NA 1.49, CFI Apo TIRF 60× H, Nikon, Japan), 485 nm laser (CUBE 485-30C, Coherent, Santa Clara, CA, USA), 642 nm laser (DL640-050, Crystalaser, Reno, NV, USA) with a rotating diffuser, an EMCCD camera (DU-897, Andor, UK) for the 485 nm channel, and an EMCCD camera (C9100, Hamamatsu, Japan) for the 642 nm channel. Synchronized image recording and flow was controlled with custom software (LabView, National Instruments, Austin, TX, USA). Shear stress in our microfluidic flow channel was proportional to pressure (100 dyn cm⁻² psi⁻¹) and reached 3000 dyn cm⁻² at 30 psi. The response time to start or stop flow in this system was fast; e.g., when flow was stopped at a shear stress of 1000 dyn cm⁻², 0.5 µm-diameter fluorescent beads (F-8813, Thermo Fisher Scientific, Waltham, MA, USA) stopped within 32 ms, close to the response time of ~10–20 ms of the solenoid valve given by the manufacturer.

**Statistical analysis.** Sample size was not predetermined. Sample sizes for each experiment were reported in corresponding figure legends. Experiments were repeated in different flow channels with similar results.

**Image analysis.** Fluorescence was analyzed with custom-written Matlab scripts. Registration mapping between Alexa Fluor 488 and Alexa Fluor 647 images used slides with multicolor 100 nm TetraSpeck microsphere (Thermo Fisher Scientific) fiducials that were excited and detected in the corresponding channels. Alexa Fluor 647 images were transformed according to the registration mapping to match the Alexa Fluor 488 images. Rectangular regions of interest (ROI) for individual VWF molecules were initially defined manually based on background-subtracted Alexa Fluor 488 images. Thresholding and particle analysis were then used to determine the contours of individual VWF molecules to refine ROIs. Length and intensity profiles of VWF and bound GPIbα were then calculated. Average intensity of single Alexa Fluor 488 or Alexa Fluor 647 fluorophores was measured by stepwise photobleaching of Alexa Fluor 488 dye-labeled streptavidin or Alexa Fluor 647 dye-labeled GPIbα sparsely attached on the glass surface. The number of VWF monomers or GPIbα molecules was determined by the total fluorescence intensity of VWF or GPIbα divided by the corresponding single Alexa Fluor dye intensity and their respective degree of labeling (0.67:1 for VWF and 1.01:1 for GPIbα). While single dye standards were attached to the surface, VWF and GPIbα bound to VWF in flow experiments could be more distal from the surface, which would decrease their fluorescence intensity in the TIRF evanescent illumination by an unknown amount, for which we did not correct. As described below, the average height of VWF was estimated to be 10–20 nm above the substrate. The average height of dyes on streptavidin were estimated to be 4.5 nm (2 × 1 nm for the two biotins +5 nm for the traptavidin/2) above the substrate. Assuming a penetration depth of 150 nm for the evanescent illumination, we estimated the ratio of illumination intensity for dyes on VWF to dyes on streptavidin to be 0.96–0.9. These estimates suggest that the correction would be small. The number of monomers in individual VWF concatemers was estimated using the total Alexa Fluor 488 fluorescence intensity of VWF divided by the degree of labeling and the intensity of single Alexa Fluor 488 fluorophores. Over 90% of the VWF concatemers maintained the same length when repeatedly elongated or held under the same shear stress. Those that did not were excluded from the data analysis. For GPIbα binding experiments, VWF concatemers that showed <20 nm average extension per monomer at 1280 dyn cm⁻², which correspond to extensions more than one SD below the mean extension (Fig. 1f), or ~16% of the total population, were excluded.

**VWF tension estimation.** The average value from two models was used to estimate the tensile force on VWF under flow. The first model simplified the VWF dimer as a dumbbell with two spheres of 13 nm radius connected by a thin tether with 94 nm length giving a length per monomer of 60 nm[52]. The tensile force on each VWF monomer induced by hydrodynamic drag force was therefore calculated

by Stoke's law

$$f_{\text{mono}} = 6\pi\mu r v \qquad (1)$$

where $\mu = 11$ cP is solution viscosity, $r = 13$ nm is the sphere radius, and $v$ is flow velocity. $v = \gamma h$, where $\gamma$ is the shear rate and $h$ is the distance between the center of VWF monomer and the flow channel surface. $h$ is estimated as $2 \times 1$ nm for the two biotins+5 nm for the traptavidin+13 nm for the radius of the dumbbell sphere = 20 nm. At 1280 dyn cm$^{-2}$, $f_{\text{mono}} = 0.6$ pN.

The second model approximates the VWF monomer as five 6 nm-diameter spheres in line with a thin 34 nm-long tether based on EM of VWF[25]. $h$ is estimated as $2 \times 1$ nm for the two biotins+5 nm for the traptavidin+3 nm for the radius of the spheres = 10 nm. At 1280 dyn cm$^{-2}$, $f_{\text{mono}} = 0.4$ pN. We used the average value of 0.5 pN from these two models to estimate the drag force on each VWF monomer at a wall shear stress of 1280 dyn cm$^{-2}$.

The tension at a specific point along a VWF concatemer was calculated by multiplying $f_{\text{mono}}$ with the total number of downstream monomers. To study the effects of VWF tensile forces on GPIα binding, 8 pN force bins from 0 to 56 pN were applied along extended VWF concatemers (Fig. 3e, upper). The number of GPIbα molecules was estimated using the total fluorescence intensity of GPIα-Alexa Fluor 647 divided by the degree of labeling and the single Alexa Fluor 647 fluorophore intensity.

**Two-state model for force-activated GPIbα binding.** Assuming that state 1 and state 2 of A1 are separated by an energy barrier along the molecular reaction axis and the free energy of state 1 is higher than state 2 by $\Delta G$, then external force at an angle $\theta$ with the reaction axis adds a mechanical potential $-f \cdot \Delta x_0 \cos\theta = -f \cdot \Delta x$, where $f$ is the external force, $\Delta x_0$ is the displacement between the two states along the molecular reaction axis, and $\Delta x$ is the displacement between the two states along the direction of the external force. Based on Boltzmann distribution, the ratio of state 1 and state 2 populations is $\exp((\Delta G - f \cdot \Delta x)/k_B T)$. The population of state 2 is thus

$$N_2 = N_{\text{total}} \cdot \frac{1}{1 + \exp((\Delta G - f \cdot \Delta x)/(k_B T))}, \qquad (2)$$

where $N_{\text{total}}$ is the total number of available A1 domains per μm of VWF concatemer, $N_2$ is the number of activated A1 domains per μm of VWF concatemer, $k_B$ is the Boltzmann constant, and $T$ is the temperature. The fractional population of state 2 is

$$P_2 = \frac{N_2}{N_{\text{total}}} = \frac{1}{1 + \exp((\Delta G - f \cdot \Delta x)/(k_B T))}. \qquad (3)$$

If state 2 has a dissociation constant $K_{D,2}$ for GPIbα binding, the number of bound GPIbα with GPIbα concentration $[C]$ as a function of force $f$ would be

$$n_{\text{GPIbα}} = \frac{[C]}{K_{D,2} + [C]} \cdot N_2 = \frac{[C]}{K_{D,2} + [C]} \cdot N_{\text{total}} \cdot P_2$$
$$= n_{\text{max}} \cdot \frac{1}{1 + \exp((\Delta G - f \cdot \Delta x)/(k_B T))}. \qquad (4)$$

Here

$$n_{\text{max}} = N_{\text{total}} \cdot \frac{[C]}{K_{D,2} + [C]}$$

If state 2 has a binding rate of $k_{\text{on},2}$ for GPIbα binding, the apparent binding rate for regions of VWF subject to tension $f$ would be

$$k_{\text{on}} = \frac{k_{\text{on},2} \cdot N_2}{N_{\text{total}}} = k_{\text{on},2} \cdot P_2 = k_{\text{on},2} \cdot \frac{1}{1 + \exp((\Delta G - f \cdot \Delta x)/(k_B T))},$$

or

$$\frac{k_{\text{on}}}{k_{\text{on},2}} = P_2 = \frac{1}{1 + \exp((\Delta G - f \cdot \Delta x)/(k_B T))}. \qquad (5)$$

**Calculation of the proposed structural change between state 1 and state 2.** In the native state of VWF A1 domain (protein databank identification 1SQ0), there are a number of hydrogen bonds external to the long-range disulfide bond (Cys$^{1458}$–Cys$^{1272}$). When A1 is in state 1, the region between Glu$^{1463}$ and Tyr$^{1271}$ is stabilized by the disulfide bond and the hydrogen bonds. The distance between Glu$^{1463}$ and Tyr$^{1271}$ is 1.2 nm. We estimated the length increase of state 2 relative to state 1 by assuming that force applied between the two termini of A1 would break the hydrogen bonds in the polypeptide segment between Glu$^{1463}$ and Cys$^{1458}$. The freed five amino acids add ~1.8 nm to the extended length of A1. The

distance between Cys$^{1458}$ and Tyr$^{1271}$ is 0.8 nm. Therefore, we estimated the net length change due to this conformational change as $1.8 + 0.8 - 1.2 = 1.4$ nm.

**VWF and GPIbα binding affinity and kinetics.** The apparent dissociation rate $k_{\text{off}}$ and association rate $k_{\text{on}}$ for GPIbα binding to regions of VWF with different tensions were determined by fitting the association curves at various GPIbα concentrations. Assuming homogeneous 1:1 binding kinetics, we have the number of bound GPIbα on VWF

$$n_{\text{GPIbα}} = N_{\text{total}} \cdot (1 - \exp(-(k_{\text{on}}[C] + k_{\text{off}})\Delta t \cdot i)) \cdot k_{\text{on}}[C]/(k_{\text{on}}[C] + k_{\text{off}}), \qquad (6)$$

where $N_{\text{total}}$ is the total number of binding sites in VWF for a certain region, $n_{\text{GPIbα}}$ is the number of bound GPIbα in the same region, $[C]$ is the concentration of free GPIbα in solution, $i$ is the frame number, and $\Delta t$ is the time lag between consecutive frames. Total concentration of GPIbα was in such great excess over $N_{\text{total}}$ that it was used for $[C]$. The dissociation constant $K_D$ was calculated as $k_{\text{off}}/k_{\text{on}}$.

**Effect of advection on A1–GPIbα dissociation kinetics.** In the experimentally studied regime, the effect of advection on the observed kinetic rates is negligible. As GPIbα binds to VWF in rapid shear flow, it is necessary to estimate the contribution of advective transport on binding kinetics. We show in the following paragraph that even at the high wall shear rate 128,000 s$^{-1}$ used in our experiment, the collision rate between GPIbα and A1 domain due to advection is two orders of magnitude lower than the one caused by diffusion. This result suggests that the contribution of advection to VWF–GPIbα binding is negligible and generally independent of flow rate.

To estimate the effect of advection on binding kinetics, we calculated and compared the collision rate between A1 and GPIbα due to diffusion and advection. The Smoluchowski diffusion-limited collision rate[53] $k_D = 4\pi R^* D_{AG}$ was calculated as the flux of ligand (GPIbα) into a spherical shell with radius $R^*$ surrounding the receptor (A1), where $R^*$ is the critical distance between A1 and GPIbα for forming an encounter complex. $R^*$ was estimated to be the sum of the radii $r$ of A1 and GPIbα. The relative diffusion coefficient between A1 and GPIbα was $D_{AG} = D_A + D_G$. The hydrodynamic radii $r$ of A1 and GPIbα were estimated to be 3 nm. Therefore,

$$D_{AG} = 2 \cdot k_B T/(6\pi\mu r), \qquad (7)$$

and

$$k_D = 4\pi \cdot 2r \cdot 2 \cdot k_B T/(6\pi\mu r) = 8k_B T/(3\mu) = 7 \times 10^9 \text{M}^{-1}\text{s}^{-1}. \qquad (8)$$

Similarly, the collision rate due to advection was calculated as the flux of GPIbα through a disk with radius $R^*$ around A1 perpendicular to the flow direction

$$k_A = \pi R^{*2} v = \pi(2r)^2 \gamma h = 9 \times 10^7 \text{M}^{-1}\text{s}^{-1}. \qquad (9)$$

where $v$ is flow velocity, $\gamma = 128,000$ s$^{-1}$ and $h$ is estimated to be 10 nm (see Methods, VWF tension estimation). Both collision rates need to be corrected by considering geometric constraints, diffusional rotation, etc. ref. [54]; however, these corrections are likely to be similar for diffusion and advection. These estimates suggest that the contribution of advection to the association rate is negligible compared to diffusion.

**Code availability.** Custom-written Matlab scripts for data analysis are available from the corresponding authors upon request, for non-profit research uses.

**Data availability.** Data that support the findings of this study are available from the corresponding authors upon request. DNA constructs will be deposited into Addgene to make available to the community.

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

## Acknowledgements

We are grateful to Baxter BioScience, Vienna, Austria for recombinant VWF. We thank Yanfeng Zhou, Mark A. Blenner, Nathan Hudson, Sha Wang, and Chafen Lu in our lab, and Mark Howarth (University of Oxford) for help with gel filtration and materials, Karen Vanhoorelbeke (Laboratory for Thrombosis Research, Belgium) and Andrew Yee (University of Michigan) for suggestions on VWF gel electrophoresis, and Benjamin Freedman (University of Washington) for discussion. We acknowledge help from the microfluidic prototyping facilities in Wyss Institute for Biologically Inspired Engineering, the Microfluidics Core Facility at Harvard Medical, and the Nikon Imaging Center at Harvard Medical School (supported by National Institutes of Health (NIH) 1S10RR026549-01). This work was supported by a National Hemophilia Foundation Judith Graham Pool Postdoctoral Research Fellowship (H.F.), National Science Foundation Graduate Research Fellowship DGE-1144152 (D.Y.), American Heart Association 13SDG17000054 (W.P.W.), NIH NIGMS R35 GM119537 (W.P.W.), the Wyss Institute (W.P.W.), NIH HL108248 (T.A.S.), and NIH HL103526 (T.A.S.).

## Author contributions

T.A.S. and W.P.W. initiated and supervised the project. T.A.S., W.P.W., H.F., Y.J. designed the research and drafted the manuscript. H.F. and Y.J. performed the experiments. Y.J., H.F., and D.Y. did data analysis. F.S. provided VWF. All authors discussed the results and commented on the manuscript.

## Additional information

**Competing interests:** The authors declare no competing financial interests.

