## [Peer Review File · Nature Communications]

Reviewers' comments:

Reviewer #1, an expert in vWF biophysics (Remarks to the Author):

Fu et al., investigated the mechanism of von Willebrand factor activation for platelet GPIIb binding. This essential first step in hemostasis is not only important in the field of VWF and hemostasis research but also for the understanding of bleeding disorders, thrombosis and stroke as well as the general behavior of polymeric proteins. The described data are therefore highly interesting to a broad audience. The study is scientifically sound and well written although at some points it could be more accessible to nonspecialists (see below). The main claim of the study is that VWF is activated by a two-step conformational transition. This statement is well substantiated by the described data and the underlying scientific question was innovatively addressed. The provided information in the methods section, including the supplemental info, should be sufficient for other researchers to reproduce the work.

There are only some minor issues that should be addressed:

Lanes 73-80, lane 152 and Figure 3: The authors state that "Measuring all regions of concatemers including bright, compact regions, or only regions of lower, more uniform fluorescence intensity gave estimates of 31 ± 12 or 58 ± 25 nm/monomer. These estimates are similar to measurements of ~ 60 and 70 nm from electron microscopy and atomic force microscopy (AFM) of intact, extended VWF or monomer fragments, respectively".

Are they really similar? In EM and AFM no forces are exerted on VWF during measurements, and timescales for immobilization of molecules onto the substrate are very likely larger than for (partial) relaxation of stretched VWF. One would thus expect to observe multimers and monomers which are incompletely stretched. In particular, the – quickly refolding – A2 domain would not be unfolded in such images. Therefore the observed 60-70 nm should be the length of incompletely stretched monomers. Since unfolding of the A2 domain adds about 30-40 nm in length (Xu and Springer, 2012), one would rather expect a length of 100nm for fully stretched monomers. In Xu and Springer, 2012, the authors described that the A2 domain unfolds at forces around 10 pN. Such forces are easily reached in the flow experiments described here. Furthermore, could it be possible that brighter signals and discrepancies in the length estimates also originate from closed stems in dimers (Zhou et al., 2011, Müller et al. PNAS 2016 etc.)? That would make estimates more difficult. The difference in the estimates should be discussed. In addition, the implications of A2 unfolding/refolding and of closed stems for the elongation and relaxation of VWF should be taken into account in the discussion.

Lanes 105-108: What are the physiological parameters that regulate elongation and relaxation? It has been described that pH and divalent cations influence intra-dimer interactions. Are the timescale of relaxation and critical shear rate for elongation dependent on pH and concentration of cations? Please also speculate on the nature of the 'attractive interactions' that accelerate relaxation.

Minor comments:

Lane 42: 'bird's nest-like conformation' leaves the impression that globular VWF always has a disk like conformation similar to red blood cells. Is there a more fitting word to describe this conformation?

Lane 59 (and Methods): How exactly was VWF purified?

Lane 130: At 10 mM NaCl VWF exhibited an increased affinity for GPIb compared to the physiological concentration of 150 mM. Please comment on the physiological relevance of this difference?

Lane 152 and Fig. 3: The fits in Figs. 3e and 3f result in different values for ΔG . This discrepancy should be discussed in the discussion section.

Lane 173-181: The authors did not mention that the force is further increased upon binding of platelets to a multimer. This aspect should be discussed.

Lane 182-192: It would be interesting to speculate on how the conformational change alters the binding interface for GPIb with respect to the proposed interaction interfaces in Blenner...Springer, 2014.

Lanes 283-284: 'incompleteness of return of VWF ... within our measurement time'. Please speculate on the reason for this incompleteness in the conclusion? Is there a hysteresis effect? Were measurements performed over longer times to check if at one point VWF completely returns to its initial state?

Lane 373-374: It should be mentioned why PCA and PCD were added to the buffer. If used as an oxygen scavenging system, the corresponding references should be added (Aitken, Marshall, and Puglisi 2008 An oxygen scavenging system for improvement of dye stability in single-molecule fluorescence experiments Biophys J 94(5):1826-35).

Lanes 480-482: The authors state that "The distance between Glu1463 and Tyr1271 is 1.2 nm. Force applied between the two termini of A1 is sufficient to break the hydrogen bonds between Glu1463 and Cys1458, which switches A1 to state 2". Was this really proven? Otherwise it should be stated as a hypothesis.

Statements/paragraphs that could be explained in a more general language to make them more accessible to nonspecialists:

Line 70: Why was the extension plotted against the log of shear stress and what physiological relevance has the value of initiation of elongation at 15 dyn/cm² ?

Lines 98-108: What is the significance of the calculated parameters?

Line 123: What "high-throughput method" was used?

Reviewer #2, an expert in shear-induced protein unfolding (Remarks to the Author):

NComms-16-2861-T Review

The paper by Springer et al. presents novel and important findings on the blood borne protein von Willebrand Factor. This important molecule is critical in the wound healing cascade. A number of studies have attempted to determine the effects of hydrodynamic forces (flow) on the unfolding

and activation of vWF. The mechanism by which the vWF acts is yet to be determined and the current paper makes significant advances in our understanding of the biophysics of VWF. The current paper combines well structured experiments using a sound knowledge of vWF biochemistry with TIRF microscopy to generate important data and new insight into the mechanism by which vWF acts. The experiments have been conducted in a rigorous manner and the interpretation is clear and logical. Generally, the paper is of the highest quality and I strongly support publication of the paper with several minor amendments.

1. How was the range of shear stresses (line 67) determined? A statement as to how these numbers were calculated should be included. Is this the stress in the fluid or that on the vWF molecule?
2. Line 79. The sizes are compared with AFM and Em values. What are the lengths of the concatomers estimated from the biochemistry?
3. Line 94. How was the viscosity increased? This should be discussed with reference to possible changes in the solvency of the vWF.
4. Line 100: What value of L_0 is used?
5. The discussion of relaxation, Lines 104-108, could have an image showing how the vWF is thought to unfold.

Reviewer #3, an expert in vWF structure and function (Remarks to the Author):

This manuscript describes data supporting a refined model for how shear force induces reversible changes in VWF conformation that promote high affinity binding of the VWF A1 domain to platelet GPIIb. In the context of the full length VWF subunit, the A1 domain has negligible affinity for platelet GPIIb under conditions of low fluid shear stress, but high affinity under conditions of high fluid shear stress. This unusual mechanism is critical for the hemostatic function of VWF, and extremely interesting as a model for rapid force-dependent metabolic regulation.

Prior studies have shown that sequences N-terminal to the A1 domain inhibit binding to GPIIb, and this inhibition can be relieved by certain mutations or by tension on the VWF subunit. However, the transition from low to high affinity state has not been studied in a way that provides insight beyond this merely qualitative description. By looking at single multimers of VWF, the authors have done a masterful job of characterizing the relationship between tension-induced conformation changes and GPIIb binding. Using dual fluorescence labels and TIRF microscopy, they obtain quantitative values for rates of multimer relaxation when flow ceases, for the forces that induce GPIIb binding, and for rates of GPIIb association and dissociation. These values clearly relate to forces that occur physiologically, and to the demands of hemostasis, as explained nicely in the discussion. This study brings us several steps closer to understanding why VWF multimers have to be so long, and precisely how VWF-dependent platelet adhesion is concentrated at sites of vascular injury.

A few comments:

The tension that activates the A1 domain is comparable to that needed to unfold the A2 domain and promote cleavage by ADAMTS13. The relationship between these two force-dependent conformational changes could be discussed.

Lines 59-63 – lightly biotinylated VWF multimers might bind have biotin in any subunit and may bind to immobilized streptavidin at any point along their length. Presumably one would prefer to look at multimers with biotin near their ends. Multimers with biotin in the middle, or immobilization through multiple biotins, would be less desirable. So how did the authors select or reject immobilized VWF multimers for the experiments of Fig. 1? Was one criterion the observation of uniform extended fluorescence intensity under flow? What fraction of immobilized VWF multimers

gave useful data? Do the authors know the approximate stoichiometry of biotin incorporation? This information could be useful to others.

Lines 71-72 – extension at about $15 \text{ dyn}\cdot\text{cm}^{-2}$ could be discussed in relation to shear stress values that occur in the human vasculature. I assume the threshold for A1 exposure is much higher when multimers are not tethered. What might it take to induce high affinity binding in solution?

Lines 113-115 – Fig 3 and Video 3 provide a remarkable, compelling demonstration that reversible tension-induced changes in VWF conformation switch the affinity for GPIb between negligible and high.

Lines 152-166 – The analysis of binding kinetics, based on the association phase and Eq. 6, appears to be valid with one potential caveat. VWF seems to elongate and expose binding sites very rapidly, compared to the time course of GPIb binding, as shown in Fig. 3b, but many things happen during elongation and exposure of the binding site in A1 might still lag compared to rapid multimer extension. If that were the case, then the rate constants would be composites and K_d values calculated from ratios would not reflect only GPIb-A1 binding. A simple check would be to compare to equilibrium K_d values determined from fluorescence intensity at long times as a function of GPIb concentration.

Lines 163-166 – A VWF monomer under no tension may have a length of about 60 nm, but at 40 pN the A2 domain is likely to unfold and roughly double the contour length of a subunit. Can the authors take this additional shear-dependent reaction into account? The two responses – A1 disinhibition and A2 unfolding – seem likely to occur under these conditions. One promotes platelet binding and the other promotes proteolysis that inhibits platelet binding. Is there a way to apply force that selectively induces one or the other response?

193-199 – The distinction between flex bond and catch bond behavior is interesting but I am not sure this explanation will be understood by readers. More to the point, the authors have not probed the effect of force on the GPIb-A1 bond in this manuscript, so I don't see how their results can speak for or against a catch bond model for GPIb-A1. Isn't exposure of the high affinity site by tension on VWF independent of the behavior of GPIb-A1 under tension?

Lines 201-203 – It is a crucial difference that prior studies have been limited to ensembles, whereas this manuscript uses single molecular methods.

Fig. 2b – cannot see the dashed lines through their entire course. Would be helpful to make them more visible.

Lines 429-431 – related to a question above. What fraction of VWF multimers were excluded? Just curious.

Lines 624-625 – Ref 15 needs page numbers, and does not seem to be an appropriate reference for the sentence at line 47.

Response to referees

We appreciate these valuable suggestions from all the reviewers. Our response to the comments is as follows. All changes have been highlighted in yellow in the manuscript.

Reviewer #1, an expert in vWF biophysics (Remarks to the Author):

Fu et al., investigated the mechanism of von Willebrand factor activation for platelet GPIb binding. This essential first step in hemostasis is not only important in the field of VWF and hemostasis research but also for the understanding of bleeding disorders, thrombosis and stroke as well as the general behavior of polymeric proteins. The described data are therefore highly interesting to a broad audience. The study is scientifically sound and well written although at some points it could be more accessible to nonspecialists (see below). The main claim of the study is that VWF is activated by a two-step conformational transition. This statement is well substantiated by the described data and the underlying scientific question was innovatively addressed. The provided information in the methods section, including the supplemental info, should be sufficient for other researchers to reproduce the work.

There are only some minor issues that should be addressed:

Comment (1):

Lanes 73-80, lane 152 and Figure 3: The authors state that “Measuring all regions of concatemers including bright, compact regions, or only regions of lower, more uniform fluorescence intensity gave estimates of 31 ± 12 or 58 ± 25 nm/monomer. These estimates are similar to measurements of ~ 60 and 70 nm from electron microscopy and atomic force microscopy (AFM) of intact, extended VWF or monomer fragments, respectively”.

Are they really similar? In EM and AFM no forces are exerted on VWF during measurements, and timescales for immobilization of molecules onto the substrate are very likely larger than for (partial) relaxation of stretched VWF. One would thus expect to observe multimers and monomers which are incompletely stretched. In particular, the – quickly refolding – A2 domain would not be unfolded in such images. Therefore the observed 60-70 nm should be the length of incompletely stretched monomers. Since unfolding of the A2 domain adds about 30-40 nm in length (Xu and Springer, 2012), one would rather expect a length of 100nm for fully stretched monomers. In Xu and Springer, 2012, the authors described that the A2 domain unfolds at forces around 10 pN. Such forces are easily reached in the flow experiments described here. Furthermore, could it be possible that brighter signals and discrepancies in the length estimates also originate from closed stems in dimers (Zhou et al., 2011, Müller et al. PNAS 2016 etc.)? That would make estimates more difficult. The difference in the estimates should be discussed. In addition, the implications of A2 unfolding/refolding and of closed stems for the elongation and relaxation of VWF should be taken into account in the discussion.

Answer (1):

This is a complex issue. We have now revised the statement in Results (Lines 87-91) to say that we are within 2-fold of previous estimates and have included a reference to Müller et al. PNAS 2016 (Reference 27). In Discussion (Lines 181-202), we go into more depth and discuss both the stem structure of Müller et al and A2 unfolding. We also found a reference (Tskhovrebova *et al*, Reference 33) that describes how surface tension at a receding meniscus during EM sample preparation combs long extensible molecules such as VWF.

Comment (2):

Lanes 105-108: What are the physiological parameters that regulate elongation and relaxation? It has been described that pH and divalent cations influence intra-dimer interactions. Are the timescale of relaxation

and critical shear rate for elongation dependent on pH and concentration of cations? Please also speculate on the nature of the ‘attractive interactions’ that accelerate relaxation.

Answer (2):

We are only concerned in this MS with behavior of VWF in flow in the vasculature relevant to hemostasis and thrombosis. While pH is important during secretion, where it rises about 2 units between the Weibel-Palade body and plasma, it varies little in the vasculature. Recently, Müller et al. (Reference 27) have described an interesting divalent-cation dependent interaction that stabilizes a stem structure in VWF dimers. However, the exact cation responsible for this interaction remains unknown, and there is little variation in cation concentration in the vasculature that could make this a regulatory system. Therefore, we have not investigated the effects of pH and divalent cations on the extension of VWF. Many investigators have attempted to identify specific domains involved in association between or within VWF concatemers. Z. Ruggeri put considerable effort into this and could not identify a specific domain. Thus far we have not, either. Association between distinct VWF concatemers could be mediated by the same interaction that mediates the fast relaxation of single VWF concatemers. A hypothesis that attracted us initially was that electrostatic interactions, perhaps between A1 and other domains, could mediate fast relaxation. This is why we tested relaxation kinetics in 10 and 150 mM NaCl. However, no effect was found, while salt has a major influence on GPIb binding. In the absence of providing any meaningful speculation on specific features involved in relaxation, we are only comfortable saying that similar interactions may mediate self-association and relaxation.

We did not have a paragraph in Discussion on relaxation and in response to this comment have now added one (Line 290-299) that clarifies what is known and not known about relaxation and self-association.

Minor comments:

Comment (3):

Lane 42: ‘bird’s nest-like conformation’ leaves the impression that globular VWF always has a disk like conformation similar to red blood cells. Is there a more fitting word to describe this conformation?

Answer (3):

Instead of “bird’s nest-like conformation”, we have now used “irregularly coiled conformation” (Line 41).

Comment (4):

Lane 59 (and Methods): How exactly was VWF purified?

Answer (4):

We have added in the first sentence in Methods (Line 314-317) all the details that have been described about the purification process, which unfortunately, remains largely proprietary.

Comment (5):

Lane 130: At 10 mM NaCl VWF exhibited an increased affinity for GPIb compared to the physiological concentration of 150 mM. Please comment on the physiological relevance of this difference?

Answer (5):

We have added the following sentence (Lines 140-142): Strong ionic dependence was consistent with the high electrostatic complementarity of the A1- GPIb α interface (Huizinga *et al*, 2002, Reference 31) and is relevant for facilitating rapid binding of GPIb α to VWF (see Discussion, Line 256-268).

Comment (6):

Lane 152 and Fig. 3: The fits in Figs. 3e and 3f result in different values for ΔG . This discrepancy should be discussed in the discussion section.

Answer (6):

It is not actually a discrepancy as the values are within error of one another. The ΔG value for on rate is less accurately determined and we have added a note to the Figure legend (Line 713-714).

Comment (7):

Lane 173-181: The authors did not mention that the force is further increased upon binding of platelets to a multimer. This aspect should be discussed.

Answer (7):

Yes, this is an important point, and we have added it to Discussion (Line 242-244, 287-289).

Comment (8):

Lane 182-192: It would be interesting to speculate on how the conformational change alters the binding interface for GPIIb with respect to the proposed interaction interfaces in Blenner...Springer, 2014.

Answer (8):

Unfortunately, the nature of the conformational change is unknown. We cannot add any meaningful Discussion- nor was it possible to do this in Blenner et al.

Comment (9):

Lanes 283-284: 'incompleteness of return of VWF ... within our measurement time'. Please speculate on the reason for this incompleteness in the conclusion? Is there a hysteresis effect? Were measurements performed over longer times to check if at one point VWF completely returns to its initial state?

Answer (9):

This statement could have been clearer. We have revised this explanation at the end of the Fig. 2 legend (Line 684-687):

“The value of 0.96 for a suggests that we can account for 96% of the relaxation with a single exponential decay. There may be another component with a slower relaxation, but we could not measure it owing to the photobleaching that would occur during the long exposure required for its measurement.”

Comment (10):

Lane 373-374: It should be mentioned why PCA und PCD were added to the buffer. If used as an oxygen scavenging system, the corresponding references should be added (Aitken, Marshall, and Puglisi 2008 An oxygen scavenging system for improvement of dye stability in single-molecule fluorescence experiments *Biophys J* 94(5):1826-35).

Answer (10):

Indeed PCA and PCD were added as an oxygen scavenging system. We have added one sentence regarding to PCA and PCD in Methods (Line 349). The corresponding reference was also cited as suggested by the reviewer.

Comment (11):

Lanes 480-482: The authors state that “The distance between Glu1463 and Tyr1271 is 1.2 nm. Force applied between the two termini of A1 is sufficient to break the hydrogen bonds between Glu1463 and Cys1458, which switches A1 to state 2”. Was this really proven? Otherwise it should be stated as a hypothesis.

Answer (11):

It was a hypothesis. We have reworded this section in Methods section “Calculation of the proposed structural change between state 1 and state 2” (Line 476-478).

Statements/paragraphs that could be explained in a more general language to make them more accessible to nonspecialists:

Comment (12):

Line 70: Why was the extension plotted against the log of shear stress and what physiological relevance has the value of initiation of elongation at 15 dyn/cm² ?

Answer (12):

One does plots such as this to find physical laws. While we found a log dependence, we do not know its physical significance, or the significance of the intercept at 15 dyn/cm², as now explained. The value of 15 dyn/cm² should also not be considered physiologically significant. We have added more wording to try to explain these issues (Line 75-82):

The observation that fractional extension vs. shear rate does not appear to depend on the length of the concatemers is difficult to explain using standard polymer models and may be dependent on specialized biochemical properties of VWF that are not currently understood. However, we did find that when extension was plotted against the log of shear stress (inset, Fig. 1d), it fit well to a straight line and gave an intercept at 15 dyn·cm⁻². This empirically observed value represents the threshold at which elongation begins, but its physiological significance may be limited, since we find that elongation is not sufficient for VWF activation, as described below.

Comment (13):

Lines 98-108: What is the significance of the calculated parameters?

Answer (13):

We have clarified their significance in the Results section (Line 107-110):

The scaling laws represented by the equations shown in Fig. 2d hold for a wide range of synthetic chemical polymers (References 29, 30). The finding that VWF follows the same laws allows prediction of the behavior of a VWF concatemer regardless of its size, and also enables comparison of VWF to other polymers.

Comment (14):

Line 123: What “high-throughput method” was used?

Answer (14):

The word “high-throughput” was not appropriate. We have deleted it from the revised manuscript.

Reviewer #2, an expert in shear-induced protein unfolding (Remarks to the Author):

NComms-16-2861-T Review

The paper by Springer et al. presents novel and important findings on the blood borne protein von Willebrand Factor. This important molecule is critical in the wound healing cascade. A number of studies have attempted to determine the effects of hydrodynamic forces (flow) on the unfolding and activation of vWF. The mechanism by which the vWF acts is yet to be determined and the current paper makes significant advances in our understanding of the biophysics of VWF. The current paper combines well structured experiments using a sound knowledge of vWF biochemistry with TIRF microscopy to generate important data and new insight into the mechanism by which vWF acts. The experiments have been conducted in a rigorous manner and the interpretation is clear and logical. Generally, the paper is of the highest quality and I strongly support publication of the paper with several minor amendments.

Comment (15):

1. How was the range of shear stresses (line 67) determined? A statement as to how these numbers were calculated should be included. Is this the stress in the fluid or that on the vWF molecule?

Answer (15):

We have added one paragraph to describe the shear stress calculation in the Methods section, under “Shear-stress control and dual-color TIRF imaging system” (Line 374-384).

Comment (16):

Line 79. The sizes are compared with AFM and Em values. What are the lengths of the concatomers estimated from the biochemistry?

Answer (16):

We have added more details on this in the first paragraph of Discussion (Page 8). One of the methods was adding up the sizes of domains in the monomer.

Comment (17):

Line 94. How was the viscosity increased? This should be discussed with reference to possible changes in the solvency of the vWF.

Answer (17):

We used 60% w/w sucrose to increase viscosity. We have added this to the Results section (Line 98-101) and the Methods section “VWF and GPIIb α ” (Line 352-353). There is unlikely to be any effect on solvation and have added a reference that suggests sucrose may increase domain stability.

Comment (18):

Line 100: What value of L_0 is used?

Answer (18):

We added the explanation of the value L_0 in Fig. 2b legend (Line 680).

The equilibrium length L_0 of individual VWF concatemers under 1280 dyn·cm⁻² flow was used.

Comment (19):

The discussion of relaxation, Lines 104-108, could have an image showing how the vWF is thought to unfold.

Answer (19):

We speculate that VWF concatemers are held in the compact structure by specific intra- and inter-monomer interactions. So far, there is still not enough evidence to let us draw a clear picture about how these interactions actually work in VWF concatemers during unfolding.

Reviewer #3, an expert in vWF structure and function (Remarks to the Author):

This manuscript describes data supporting a refined model for how shear force induces reversible changes in VWF conformation that promote high affinity binding of the VWF A1 domain to platelet GPIb. In the context of the full length VWF subunit, the A1 domain has negligible affinity for platelet GPIb under conditions of low fluid shear stress, but high affinity under conditions of high fluid shear stress. This unusual mechanism is critical for the hemostatic function of VWF, and extremely interesting as a model for rapid force-dependent metabolic regulation.

Prior studies have shown that sequences N-terminal to the A1 domain inhibit binding to GPIb, and this inhibition can be relieved by certain mutations or by tension on the VWF subunit. However, the transition from low to high affinity state has not been studied in a way that provides insight beyond this merely qualitative description. By looking at single multimers of VWF, the authors have done a masterful job of

characterizing the relationship between tension-induced conformation changes and GPIb binding. Using dual fluorescence labels and TIRF microscopy, they obtain quantitative values for rates of multimer relaxation when flow ceases, for the forces that induce GPIb binding, and for rates of GPIb association and dissociation. These values clearly relate to forces that occur physiologically, and to the demands of hemostasis, as explained nicely in the discussion. This study brings us several steps closer to understanding why VWF multimers have to be so long, and precisely how VWF-dependent platelet adhesion is concentrated at sites of vascular injury.

A few comments:

Comment (20):

The tension that activates the A1 domain is comparable to that needed to unfold the A2 domain and promote cleavage by ADAMTS13. The relationship between these two force-dependent conformational changes could be discussed.

Answer (20):

Correct; we have added a paragraph on this topic to the Discussion (Line 238-245).

Comment (21):

Lines 59-63 – lightly biotinylated VWF multimers might bind have biotin in any subunit and may bind to immobilized streptavidin at any point along their length. Presumably one would prefer to look at multimers with biotin near their ends. Multimers with biotin in the middle, or immobilization through multiple biotins, would be less desirable. So how did the authors select or reject immobilized VWF multimers for the experiments of Fig. 1? Was one criterion the observation of uniform extended fluorescence intensity under flow? What fraction of immobilized VWF multimers gave useful data? Do

the authors know the approximate stoichiometry of biotin incorporation? This information could be useful to others.

Answer (21):

We cannot distinguish the immobilization location of VWF concatemers and therefore are presenting data from VWF concatemers that were immobilized at all possible locations along the length. The intensity profile is not a reliable criterion for the immobilization location because the non-uniform intensity profile observed in our experiment might be due to a number of other reasons. We have now explicitly described the stoichiometry of biotin labeling and the issue of where binding occurs along the length of a concatemer in the first paragraph of Results (Line 59-66).

Comment (22):

Lines 71-72 – extension at about 15 dyn•cm⁻² could be discussed in relation to shear stress values that occur in the human vasculature. I assume the threshold for A1 exposure is much higher when multimers are not tethered. What might it take to induce high affinity binding in solution?

Answer (22):

A few sentences have been added to the Results section “VWF elongation and relaxation in flow” (Line 75-82) as described above in response to Comment (12). Furthermore, we have added a paragraph on this topic in Discussion (Lines 269-279).

Comment (23):

Lines 113-115 – Fig 3 and Video 3 provide a remarkable, compelling demonstration that reversible tension-induced changes in VWF conformation switch the affinity for GPIb between negligible and high.

Thanks for the comment.

Comment (24):

Lines 152-166 – The analysis of binding kinetics, based on the association phase and Eq. 6, appears to be valid with one potential caveat. VWF seems to elongate and expose binding sites very rapidly, compared to the time course of GPIb binding, as shown in Fig. 3b, but many things happen during elongation and exposure of the binding site in A1 might still lag compared to rapid multimer extension. If that were the case, then the rate constants would be composites and K_D values calculated from ratios would not reflect only GPIb-A1 binding. A simple check would be to compare to equilibrium K_D values determined from fluorescence intensity at long times as a function of GPIb concentration.

Answer (24):

It is a very good point that the association rate of GPIb α we measured can be composite. However, from the data we have, we are unable to separate the association process after flow stretching VWF. The initial slope of the intensity trace is a sum of both the on and off rates (see Eq. 6). To obtain the on rate, we are already using the equilibrium K_D values determined from fluorescent intensity at long times by fitting Eq. 6 to intensity traces obtained at a range of GPIb α concentration (Supplementary Fig. 5). If K_D is calculated from fluorescence intensity at long times only as a function of GPIb α concentration, the value is 71 ± 76 nM. Obviously the uncertainty in this measure is high, however, even without including the error it agrees well with the value determined by fitting the entire traces. The referee may also check the fit curves in Supplementary Fig. 5. They show that even when rates are included in the fitting, the fit agrees well with the amount of binding at the latest time points, i.e. the match to maximum binding near equilibrium is good.

Ideally, the off rates would be measured independently by removing GPIb α from the flow stream. However, we are currently technically limited in our ability to do so while maintaining a stable high flow rate. A microfluidic device that is capable of exchanging buffer without disruption the high flow rate may be developed in future to address this issue.

Comment (25):

Lines 163-166 – A VWF monomer under no tension may have a length of about 60 nm, but at 40 pN the A2 domain is likely to unfold and roughly double the contour length of a subunit. Can the authors take this additional shear-dependent reaction into account? The two responses – A1 disinhibition and A2 unfolding – seem likely to occur under these conditions. One promotes platelet binding and the other promotes proteolysis that inhibits platelet binding. Is there a way to apply force that selectively induces one or the other response?

Answer (25):

Paragraphs have been added to the Discussion section to discuss A2 unfolding (Lines 238-245). One cannot selectively induce one or another with WT VWF. It would certainly be an interesting experiment.

Comment (26):

193-199 – The distinction between flex bond and catch bond behavior is interesting but I am not sure this explanation will be understood by readers. More to the point, the authors have not probed the effect of force on the GPIb-A1 bond in this manuscript, so I don't see how their results can speak for or against a catch bond model for GPIb-A1. Isn't exposure of the high affinity site by tension on VWF independent of the behavior of GPIb-A1 under tension?

Answer (26):

Thanks for the comment. We have clarified this issue in the Discussion (Line 224-237). We are trying to relate our results to existing models.

Comment (27):

Lines 201-203 – It is a crucial difference that prior studies have been limited to ensembles, whereas this manuscript uses single molecular methods.

Thank you for the comment.

Comment (28):

Fig. 2b – cannot see the dashed lines through their entire course. Would be helpful to make them more visible.

Answer (28):

We apologize for the invisibility of the dashed lines due to over 200 lines overlapping with each other. A new Supplementary Figure was added in our manuscript with the solid and dashed lines in separate panels (see Supplementary Fig. 1).

Comment (29):

Lines 429-431 – related to a question above. What fraction of VWF multimers were excluded? Just curious.

Answer (29):

For VWF extension data, <10% of the total population was excluded. For VWF/GPIIb/IIIa binding data, about 16% of the total population was excluded, as explained in Methods (Line 422-426).

Comment (30):

Lines 624-625 – Ref 15 needs page numbers, and does not seem to be an appropriate reference for the sentence at line 47.

Answer (30):

We have added the page number 5030-5042 (Line 560). On pages 5033-5035, the reference discusses the rapid periodic transition between the coiled and stretch state under Couette flow (shear flow) and the stable stretch state under longitudinal gradients (elongational flow). It seems to be an appropriate reference supporting our sentence.

Additional point. Besides these changes, we found that the number of bound GPIIb α we presented was off by a factor of 2.3 because we made an error in the measurement of single Alexa 647 dye intensity. These numbers and the interpretation have been corrected in the updated manuscript. These changes include (1) the average number of GPIIb α per μm of VWF (the y axis in Fig. 3e), (2) the values of $N_{total}/\mu\text{m}$ and nm/binding site at tension 40-56 pN for both 10 and 150 mM NaCl in Fig. 3g, (3) Line 173-174 at the end of Results, and (4) Supplementary Fig. 5.

REVIEWERS' COMMENTS:

Reviewer #1 (Remarks to the Author):

All my concerns have appropriately been addressed.
Very nice work!

Reviewer #3 (Remarks to the Author):

The revisions have addressed my concerns. One new point based on new text:

Page 12, lines 287-289 – A couple of other papers have reported that platelet binding to VWF increases the force transmitted to VWF at a given shear rate, and thereby increases cleavage by ADAMTS13. For example, see Shim et al 2008 (PMID: 17901248) and Skipworth et al 2010 (PMID: 20605782). One or both of these papers might be relevant here, or perhaps at lines 242-244.

Response to referees

We appreciate all the reviewers for the valuable suggestions. Our response to the comments is as follows.

Reviewer #1 (Remarks to the Author):

Comment: All my concerns have appropriately been addressed. Very nice work!

Thank you very much.

Reviewer #3 (Remarks to the Author):

The revisions have addressed my concerns. One new point based on new text:

Comment:

Page 12, lines 287-289 – A couple of other papers have reported that platelet binding to VWF increases the force transmitted to VWF at a given shear rate, and thereby increases cleavage by ADAMTS13. For example, see Shim et al 2008 (PMID: 17901248) and Skipworth et al 2010 (PMID: 20605782). One or both of these papers might be relevant here, or perhaps at lines 242-244.

Answer: We have cited both papers in Lines 244 and 288.